# Glycolaldehyde induces synergistic effects on vascular inflammation in TNF-α-stimulated vascular smooth muscle cells

**Hee-Weon Lee**[1], **Min Ji Gu**[1], **Guijae Yoo**[1], **In-Wook Choi**[1], **Sang-Hoon Lee**[1,2], **Yoonsook Kim**[1], **Sang Keun Ha**[1,2] *

**1** Korea Food Research Institute, Wanju-gun, Republic of Korea, **2** Division of Food Biotechnology, University of Science and Technology, Daejeon, Korea

* skha@kfri.re.kr

**Data Availability Statement:** All relevant data are within the paper and its Supporting Information files.

## Abstract

Atherosclerosis is a chronic inflammatory disease that contributes to disease progression is associated with the expression of adhesion molecules in vascular smooth muscle cells (VSMCs). Glycolaldehyde (GA) has been shown to impair cellular function in various disorders, including diabetes, and renal diseases. This study investigated the effect of GA on the expression of adhesion molecules in the mouse VSMC line, MOVAS-1. Co-incubation of VSMCs with GA (25–50 μM) dose-dependently increased the protein and mRNA level of Vcam-1 and ICAM-1. Additionally, GA upregulated intracellular ROS production and phosphorylation of MAPK and NK-κB. GA also elevated TNF-α-induced PI3K-AKT activation. Furthermore, GA enhanced TNF-α-activated IκBα kinase activation, subsequent IκBα degradation, and nuclear translocation of NF-κB. These findings suggest that GA stumulated VSMC adhesive capacity and the induction of VCAM-1 and ICAM-1 in VSMCs through inhibition of MAPK and NF-κB signaling pathways, providing insights into the effect of GA to induce inflammation within atherosclerotic lesions.

## Introduction

Atherosclerosis is a chronic inflammatory disorder characterized by accumulation of lipids and recruitment of leukocytes in arterial vessels. These phenomena are similar in other vascular diseases such as cardiovascular disease [1, 2]. As arteriosclerosis progresses, vascular smooth muscle cells (VSMCs) physically interact with inflammatory leukocytes; this is an essential factor in the occurrence and exacerbation of the disease. It also indicates that VSMCs have an essential role in the progression of atherosclerosis [3–5]. The cellular adhesion molecules (CAM), such as vascular cell adhesion molecule-1 (VCAM-1) and intracellular adhesion molecule-1 (ICAM-1) in blood vessels have a critical role in the progression of lesions in atherosclerosis [6]. In the early stage of atherosclerosis, the expression of VCAM-1 and ICAM-1 is upregulated as the inflammatory response increases, and the increased expression of CAM promotes the accumulation of inflammatory leukocytes in the vascular endothelium [6–8]. Furthermore, as atherosclerosis progresses, inflammatory cytokines are secreted from

**Funding:** This research was supported by the Main Research Program (E 0210200) of the Korea Food Research Institute (KFRI) funded by the Ministry of Science and National Research Foundation of Korea (NRF) grant funded by the Korea government(MSIT) (NRF-2020R1A2C2012608). The fundes have role in conceptualization, formal analysis, investigation, project administration, and writing of the manuscript.

**Competing interests:** The authors have declared that no competing interests exist.

**Abbreviations:** AGE, advanced glycation end products; GA, glycolaldehyde; ICAM-1, intercellular adhesion molecule-1; IκBα, nuclear factor of kappa light polypeptide gene enhancer in B-cells inhibitor alpha; IL-6, interleukin-6; MAPKs, mitogen-activated protein kinase; NF-κB, nuclear factor-kappa B; PI3K, Phosphoinositide 3-kinase; RAGE, receptor for advanced glycation end products; TNF-α, Tumor necrosis factor alpha; VCAM-1, vascular cell adhesion molecule-1; VSMC, Vascular smooth muscle cell.

endothelial cells and phagocytes, which further stimulate the inflammatory responses [9, 10]. Based on these findings, the regulation of CAM expression on VSMCs is critical for the control of lesions. In addition, factors related to the expression of CAM may control the inflammatory process in VSMCs.

Advanced glycation end products (AGEs) known as glycotoxins are oxidizing compounds that are pathogenic in chronic inflammatory disorders including diabetes and atherosclerosis [11]. In the metabolic process, the generation of AGE is a normal phenomenon. However, excessive production and accumulation of AGEs are toxic to the organisms [12]. AGEs directly induces the secretion of various cytokines, hormones and free radicals in cells, resulting in cell thickening, infiltration of inflammatory cells and accumulation of extracellular matrix [13, 14]. The deposition of AGEs is involved in the activation of inflammatory cytokines and exacerbation of atherosclerosis in arterial vessels [15]. At the same time, it causes the initiation and generation of oxidative stress through the production of oxygen free radicals. However, the process of AGE accumulation and lesion progression are only partially understood in vascular disease. Likewise, it is not clear how to control the disease. Damage to VSMCs caused by AGEs has been shown to be mediated by inflammatory responses and ROS, suggesting that this is one of the mechanisms by which AGEs may alter the function of VSMCs [16]. Therefore, precursors that play an important role in AGE production are an important part of this mechanism. The AGE-RAGE axis is known to induce cellular oxidative stress in endothelial cells. Typically, low-molecular-weight carbonyl compounds such as methylglyoxal (MGO) and glyoxal (GO) are formed under hyperglycemic conditions and act as precursors to AGE [17, 18]. They also form adducts on proteins, leading to cellular dysfunction associated with complications of diabetes. Therefore, we tried to determine the effect of GA-induced AGE in the atherosclerotic environment using GA, which acts as a precursor to AGE production [19].

Receptor for advanced glycation end products (RAGE) is a member of the immunoglobulin superfamily and is expressed on VSMCs and endothelial cells [20]. In atherosclerosis, the binding of AGE to RAGE is enhanced in vascular cells and causes increased oxidant stress in the vascular wall [21]. The binding of AGE to RAGE stimulates the signaling pathways involving mitogen-activated protein kinases (MAPKs) and nuclear factor-kappa B (NF-kB), and induces oxidative stress by increasing of ROS generation, inducing various cellular responses [22, 23]. In addition, accumulated evidences showed that oxidative stress through the AGE-AGE axis induces an inflammatory responses of blood vessels in atherosclerosis [24]. However, the effects of the inflammatory responses induced by the precursors of AGE are less well understood, and mechanisms causing inflammatory reactions have not been studied. In addition, many studies have been studied to confirm the effects of various AGEs, but studies using GA precursors have not been conducted. We confirmed how synergistic effect of GA, a precursor of AGE, was in the cellular environment of atherosclerosis-induced conditions. Therefore, this study determined to investigate the mechanisms and synergistic effects of action of the AGE precursor glycolaldehyde (GA) in CAM accumulation after TNF-α treatment.

## Materials and methods

### Chemical reagents and antibodies

GA (23147-58-2) and aminoguanidine (AG, 1937-19-5) were purchased from Sigma-Aldrich (St. Louis, MO, USA). Dulbecco's modified Eagle's medium (DMEM), and fetal bovine serum (FBS) were obtained from Gibco (BRL, Carlsbad, CA, USA). Reporter plasmid pGL3-NF-κB and pCMV-β-gal used in the luciferase assay system were obtained from Promega (Madison, WI, USA). Most chemicals, including MAPK inhibitors, were obtained from Sigma Chemical Co. (St. Louis, MO, USA) unless otherwise stated. Antibodies against target molecules were

obtained from Cell signaling (Danvers, MA, USA), LSBio (Seattle, WA, USA), and Santa Cruz Biotechnology (Santa Cruz, CA, USA) unless otherwise stated. The following antibodies were used: AGEs (LS-C664030) from LSBio (Seattle, WA, USA), VCAM-1 (#39036), RAGE (#42544), phospho-p65 (#3033), p65 (#8242), phospho-IκB (#2859), IκB (#9242); all from Cell Signaling Technologies (Danvers, MA, USA), ICAM-1 (sc8439), phospho-ERK (sc81492), ERK (sc7383), phospho-JNK (sc81502), JNK (sc6254), phospho-p38 (sc166182), p38 (sc271120), phospho-PI3K (sc166365), phospho-AKT (sc514032), TNF-α (sc52746), IL-6 (ab57315); all from Santa Cruz Biotechnology (Santa Cruz, CA, USA).

## Cell culture

The mouse VSMC line MOVAS-1 was purchased from ATCC (Rockville, MD, USA) and grown in DMEM supplemented with 10% heat-inactivated FBS. The cells were incubated at 37˚C in a humidified incubator containing 5% $CO_2$ and sub-cultured once every two days.

Human aorta VSMC (HA-VSMC) was purchased from ATCC (Rockville, MD, USA) and grown in DMEM high-glucose medium with L-glutamine (PAA Laboratories, Pasching, Austria) containing 10% fetal bovine serum (Biochrom) at 37˚C and 5% $CO_2$ and sub-cultured once every three or four days. After GA or TNF-α (Sigma, 94948-59-1) treatment (dilution for use), the cells were lysed with homogenization buffer.

## Assessment of cell proliferation

This technique uses the principle that mitochondria reduce MTT to insoluble formazan. Cell proliferation was investigated for about 2 h using the MTT quantitative colorimetric assay to detect the mitochondrial activity in living cells. Cell proliferation was investigated for about 2 h using the MTT quantitative colorimetric assay to detect the mitochondrial activity in living cells. The absorbance was measured using an ELISA reader (Molecular Devices, Carlsbad, CA, USA) at 540 nm.

## ROS production assay

ROS production was quantified by fluorescence microscopy (ZEISS, Oberkochen, Germany) using a 2′,7′-dichlorofluorescein diacetate probe (DCF-DA). Mouse VSMCs were incubated with 10 μM of DCF-DA under dark conditions for 30 min at 37˚C, and rinsed with phosphate-buffered saline (PBS). ROS production was measured using an ELISA plate reader (Molecular Devices) at 488 nm excitation and 522 nm emission wavelengths.

## Immunoblotting

Cells were stimulated with different concentrations of GA (25–50 μM). After stimulation, the cultured cells were rinsed in PBS and suspended in a homogenizer lysis buffer comprised of 0.1% sodium dodecyl sulfate, 0.5% sodium deoxycholate, 1% NP-40, 1 μg/ml pepstatin, 2 μg/ml aprotinin, 10 μg/ml leupeptin and 100 μg/ml phenylsulfonyl fluoride and 150 mM NaCl in 50 mM Tris, pH 8.0. The protein concentration was determined using a DC protein assay kit (Bio-Rad, Hercules, CA, USA) with bovine serum albumin as the standard. The whole cell lysates were seperated by 6–15% SDS-PAGE and transferred to nitrocellulose membranes (Bio-Rad). The membranes were blocked with 5% skim milk in TBST (Mixture of Tris-buffered saline and Polysorbate 20) at 20–22˚C for 1 h, and probed with the appropriate primary (1:500) and secondary (1:5000) antibodies. These blots were developed using an enhanced chemiluminescence kit (DOGEN, Seoul, Korea).

## Cytosol and nuclear extract preparation

Nuclear/Cytosol fractionation kit (ab289882) was purchased from abcam (Cambridge, UK). The cultured cells were pelleted by centrifugation, and then rinsed 2–3 times in iced PBS. Pelleted cells were resuspended in buffer A (10 mM HEPES, 1.5 mM MgCl2, 10 mM KCl, 0.5 mM DTT, 0.05% NP40 (or 0.05% Igepal or Tergitol) pH 7.9) and incubated on ice for 1 h with vortexing. Subsequently, cytosol extract in the supernatant was obtained by centrifugation. The nuclear protein pelleted by centrifugation was suspended in Buffer C (5 mM HEPES, 1.5 mM MgCl2, 0.2 mM EDTA, 0.5 mM DTT, 26% glycerol (v/v), pH 7.9) and incubated on ice for 1 h with vortexing every 15 min. The supernatant containing the nuclear protein extract is separated by centrifugation and transferred to a new centrifuge tube to obtain pure nuclear protein. Separated cytoplasmic and nuclear proteins were stored at -20˚C.

## Quantitative real-time polymerase chain reaction (qRT-PCR)

Cells were stimulated in the presence or absence of GA (25–50 μg/mL) for 24 h. After GA treatment, total RNA was isolated from cultured cells using RNA extraction kit (Kusatsu, St. Shiga, Japan) and used for cDNA synthesis (Bio-Rad, Hercules, CA, USA). After cDNA synthesis, 10 μL of SYBR green premix (BioRad), 8 μL of sterile water, and 1 μL each of forward and reverse primer were mixed to obtain the total volume of 20 μL. Fluorescence was measured at each cycle. The RT-PCR primer sequences used to examine the expression of cytokines are indicated in Table 1.

## Immunofluorescence microscopy

The expression of NF-κB proteins in GA-stimulated cells was determined by immunofluorescence microscopy. VSMCs were rinsed in PBS and fixed with 3.7% formaldehyde in PBS for 30 min at 20–22˚C. The cells were permeabilized with 0.2% Triton X-100 in PBS for 1 h and then, incubated with antibodies against NF-κB p65 overnight at 4˚C. Following PBS washing, the cells were incubated for 1 h with anti-rabbit IgG-fluorescein isothiocyanate (FITC) in PBS with 0.2% Triton X-100. The samples were photographed using an LSM 900 fluorescence microscope (ZEISS).

## Statistical analyses

Each result is reported as mean ± S.E.M. One-way analysis of variance (ANOVA) was used to determine significance among the groups, after which the modified $t$–test and two-way ANOVA were used for comparison between individual groups. Significant values ($p < 0.05$) are represented by an asterisk.

**Table 1. Primer sequences and real-time PCR conditions.**

| Gene | Forward primer (5' → 3') | Reverse primer (5' → 3') |
|------|--------------------------|--------------------------|
| Vcam-1 | CCC AAG GAT CCA GAG ATT CA | TAA GGT GAG GGT GGC ATT TC |
| Icam-1 | CCT GTT TCC TGC CTC TGA AG | GTC TGC TGA GAC CCC TCT TG |
| RAGE | AGG AGG AAG AGG AGG AGC GT | TGG CAA GGT GGG GTT ATA CAG |
| TNF- | CCC TCA CAC TCA GAT CAT CTT CT | GCT ACG ACG TGG GCT ACA G |
| IL-6 | CCA CGG CCT TCC CTA CTT C | TTG GGA GTG GTA TCC TCT GTG A |
| GAPDH | TGC ATC CTG CAC CAC CAA | TCC ACG ATG CCA AAG TTG TC |

## Results

### GA induces TNF-α-induced vascular cell adhesion molecule expression

To determine whether GA regulates the CAM protein VCAM-1 and ICAM-1, VSMCs were incubated with various concentrations of GA (25–50 μM) under TNF-α (10 ng/mL) treatment. TNF-α upregulated the expression of CAM in stimulated MOVAS-1 cells. In addition, GA treatment significantly upregulated the expression of TNF-stimulated CAM expression in a concentration-dependent manner (Fig 1A and 1B). In particular, it was confirmed that the expression of CAM proteins was markedly increased when treated with 50 μM of GA. We also elucidated the mRNA level of adhesion molecules by qRT-PCR analysis. VSMCs were pre-treated with various GA concentrations (25–50 μM) in the presence of TNF-α for the same durations as previously described. As shown in Fig 1C and 1D, GA treated VSMCs had markedly upregulated VCAM-1 and ICAM-1 mRNA levels, which was comparable to their protein expression. To confirm whether GA regulate adhesion molecules in TNF-α-treated HA-VSMCs, human primary cells, we investigated the effects of GA on TNF-α-induced CAM proteins in HA-VSMCs (Fig 1E and 1F). GA also significantly increased adhesion molecules expression in primary cells. Taken together, these data demonstrated remarkable induction of CAM expression by GA in TNF-α-activated VSMCs, indicating that GA is a cause of deteriorating VSMCs by regulating the mRNA and protein levels of the CAM proteins.

### GA induces TNF-α-induced AGEs and RAGE expression

To evaluate the effect of GA on the formation of AG in TNF-α treated VSMCs, we investigated the protein level of AGE by western blot analysis. VSMCs were treated with GA at 25–50 μM for 24 h in the presence of TNF-α. GA remarkably increased the production of AGE in a concentration-dependent manner. However, when AG, an AGE inhibitor, was added, AGE production was suppressed when compared with GA treatment at 50 μM (Fig 2A). We also assessed the effect of GA on the RAGE protein and mRNA expression was increased in a concentration-dependent manner in VSMCs. VSMCs were incubated to GA at various concentrations (25–50 μM) for 24 h, which increased protein and mRNA expression of RAGE in VSMCs in a concentration-dependent manner. In contrast, AG, an agent that inhibits AGE, decreased RAGE protein and mRNA levels (Fig 2B and 2C). Our data showed that GA increases AGE and RAGE production. Furthermore, these findings revealed that the production of RAGE by GA plays a key role in VSMCs.

### GA increases TNF-α-induced activation of NF-κB

It is well known that NF-κB is a transcription factor that has a critical role in the inflammatory response in chronic inflammatory diseases. Further, NF-κB acts as an important mediator of adhesion molecule expression. Interestingly, NF-kB contains the promoter of VCAM-1 and plays a critical role in the inflammatory response. Therefore, we elucidated the effect of GA on the activation of NF-kB. The MOVAS-1 cells were incubated with different GA concentrations in the presence of TNF-α for 4 h. As shown in Fig 3A and 3B, western blot analysis exhibited that GA stimulated NF-κB p65 translocation from cytosol to the nucleus and increased IκBα phosphorylation. In contrast, the increased levels of NF-κB p65 and IκBα were significantly suppressed by pre-incubation with AG followed by GA treatment. Furthermore, immunofluorescence microscopy revealed that nuclear translocation of p65 subunits was accelerated by GA in TNF-α-activated MOVAS-1 cells (Fig 3C). Collectively, our data suggested that GA stimulates NF-κB activation through IκBα proteolytic degradation and phosphorylation of NF-κB subunits.

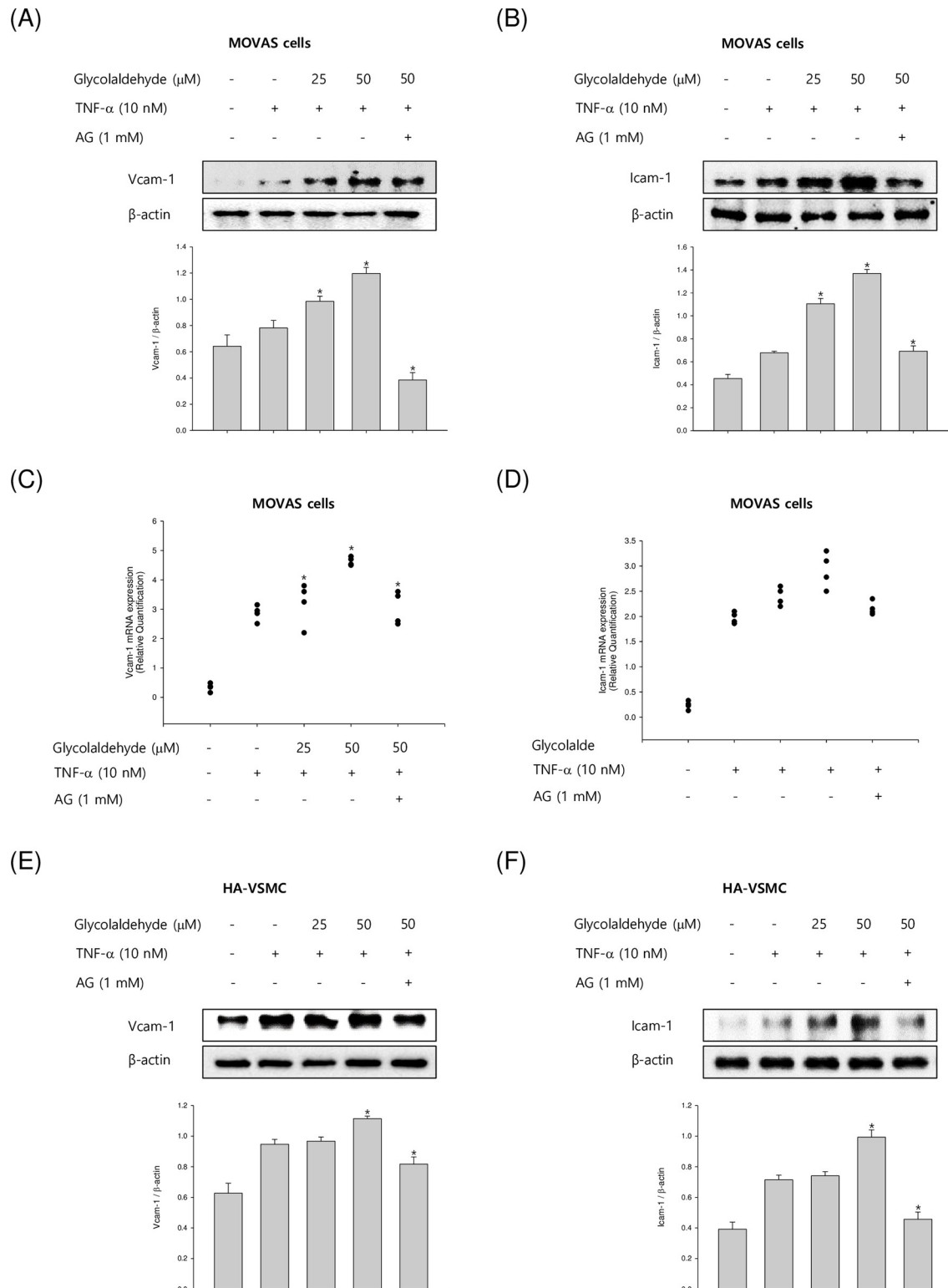

**Fig 1. Effects of GA on TNF-α-stimulated adhesion molecule protein and mRNA levels in VSMCs.** (A and B) The mouse VSMCs, MOVAS-1 cells, were incubated with TNF-α (10 ng/mL) in the presence or absence of GA (25 and 50 μM) for 8 h. The CAM protein levels of whole cells were investigated by western blotting. (C and D) MOVAS-1 cells were stimulated with TNF-α (10 ng/mL) in the presence or absence of GA (25 and 50 μM) for 4 h. The mRNA level of VCAM-1 and ICAM-1 was investigated by qRT-PCR. GAPDH served as the internal control. (E and F) The human primary cells, HA-VSMCs, were incubated with TNF-α (10 ng/mL) in

the presence or absence of GA (25 and 50 μM) for 24 h. Results are shown as means ± SEM from a representative experiment (n = 5).
*p<0.05 significantly different from the group treated with TNF-α.

## Effect of GA on MAPKs in TNF-α-stimulated VSMCs

Previous studies reported that treatment of TNF-α increases the activation of MAPK, thereby increasing the expression of CAM. In this study, we demonstrated that treatment of GA further stimulates TNF-α-activated cells, affecting CAM expression. Therefore, we investigated whether the induction effect of GA on CAM formation was dependent on the MAPKs pathway. Fig 4A shows the levels of ERK1/2, JNK, and p38 MAPK in the TNF-α-activated cells. In addition, MAPK phosphorylation was increased following GA treatment. To confirm whether

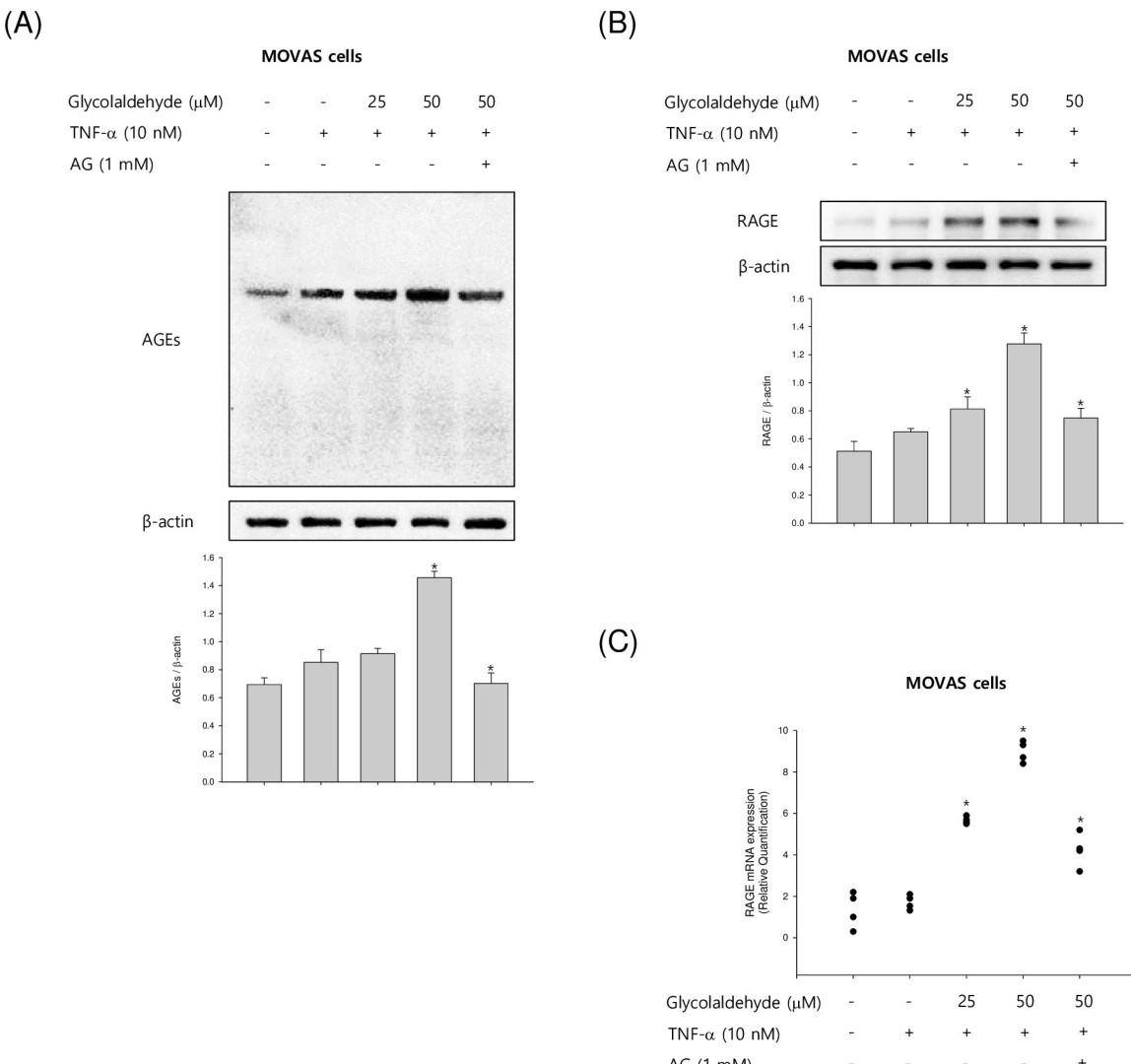

**Fig 2. Effects of GA on the expression of AGE and RAGE in TNF-α-stimulated VSMCs.** (A and B) The mouse VSMCs, MOVAS-1 cells, were incubated with TNF-α (10 ng/mL) in the presence or absence of GA (25 and 50 μM) for 24 h. (B) MOVAS-1 cells were stimulated with TNF-α (10 ng/mL) in the presence or absence of GA (25 and 50 μM) for 4 h. The whole cell lysates were investigated by western blot assay. Results are shown as means ± SEM from a representative experiment (n = 5). *p<0.05 significantly different from the group treated with TNF-α.

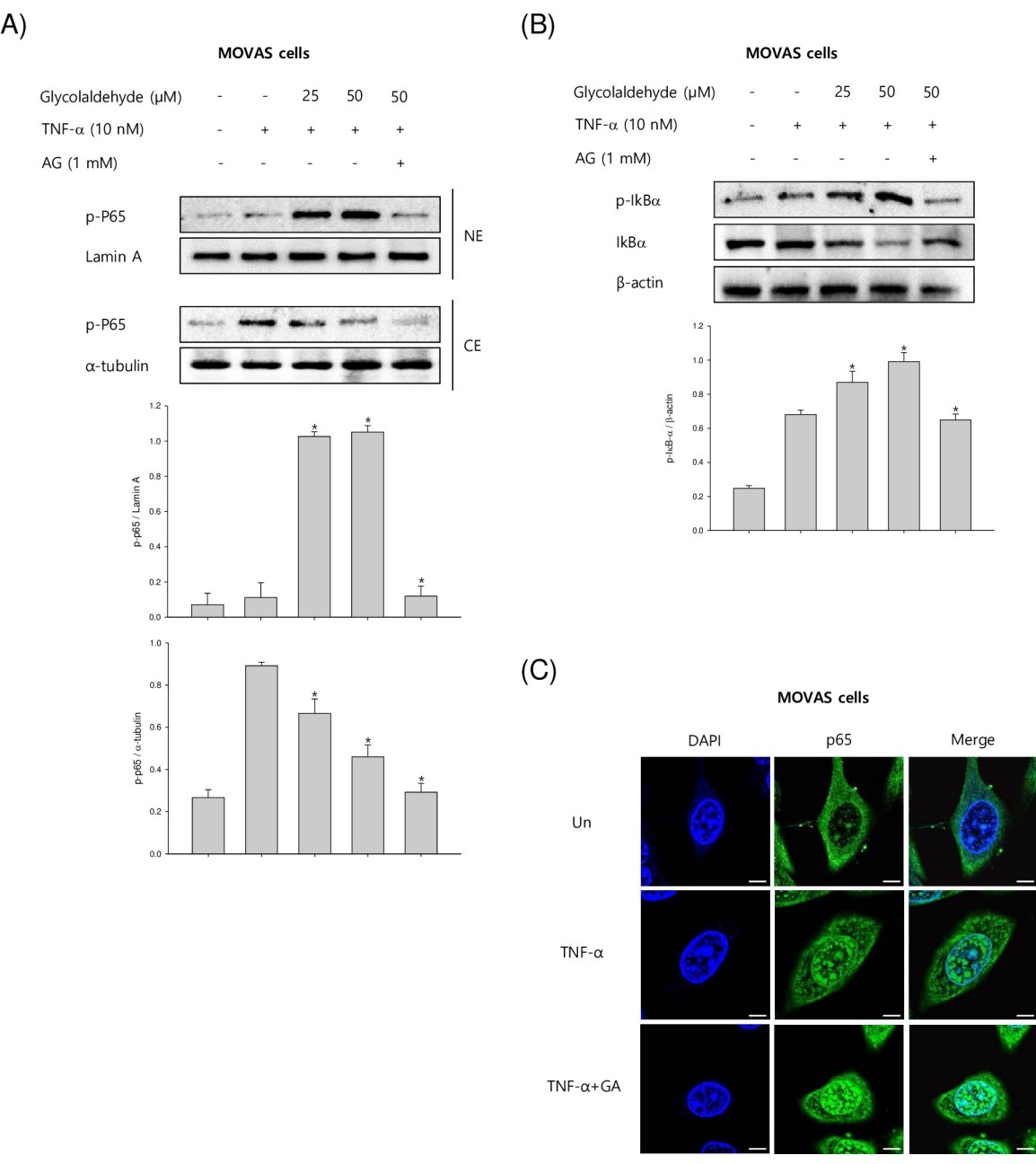

**Fig 3. Effects of GA on TNF-α-induced activation and translocation of NF-κB in VSMCs.** (A, B) The mouse VSMCs, MOVAS-1 cells, were stimulated with TNF-α (10 ng/mL) in the presence or absence of GA (25 and 50 μM) for 4 h. The nuclear protein levels of p65 and IκBα were identified by Western blot assay to demonstrate the translocation of NF-κB p65. (C) MOVAS-1 cells were stimulated with TNF-α (10 ng/mL) in the presence or absence of GA (25 and 50 μM) for 4 h. After stimulation, the cells were incubated with the NF-κB p65 primary antibody followed by FITC-labeled anti-rabbit IgG antibody. The cells were observed using fluorescence microscopy at 600× magnification. The level of lamia A and α-tubulin was measured for nuclear and cytosol as an internal control, respectively. Results are shown as means ± SEM from a representative experiment (n = 5). *$p<0.05$ significantly different from the group treated with TNF-α.

MAPK pathways regulate adhesion molecules and inflammatory cytokine production in TNF-α-treated MOVAS-1 cells, we investigated the effects of MAPK inhibitors, pathway-specific inhibitors, on TNF-α-induced CAM and inflammatory cytokine secretion. VSMCs were pre-treated with inhibitors for 2 h before TNF-α exposure and found inhibitory effects on the

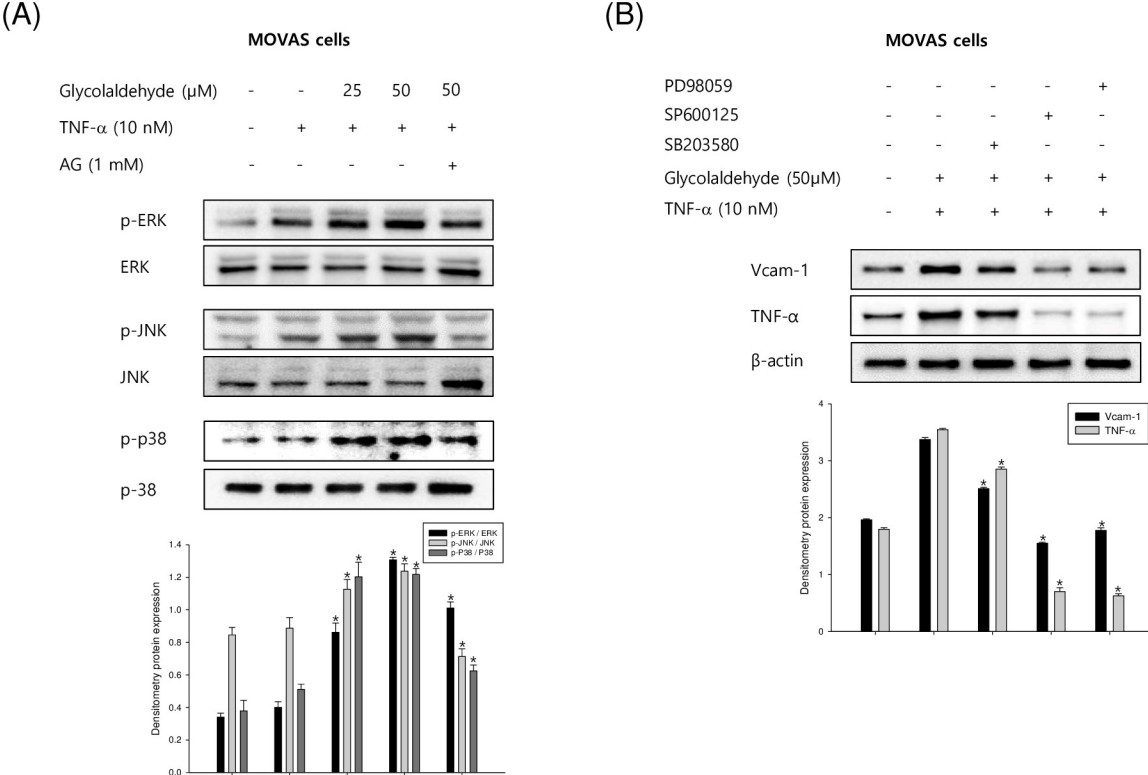

**Fig 4. Effects of GA on the phosphorylation of MAPKs in TNF-α-stimulated VSMCs.** (A) The mouse VSMCs, MOVAS-1 cells, were incubated with TNF-α (10 ng/mL) in the presence or absence of GA (25 and 50 μM) for 30 min. (B) MOVAS-1 cells were incubated with GA and TNF-α (10 ng/mL) in the presence or absence of the ERK1/2 inhibitor PD98059 (20 μM), the JNK inhibitor SP600125 (10 μM), and the p38 MAPK inhibitor SB203580 (10 μM). MAPKs protein levels were determined by western blot assay. Results are shown as means ± SEM from a representative experiment (n = 5). *$p < 0.05$ significantly different from the group treated with TNF-α.

TNF-α-activated adhesion molecules and inflammatory cytokine levels (Fig 4B). Among MAPK pathways, TNF-α-induced phosphorylation of ERK1/2 showed the greatest attenuation following inhibitor pretreatment. The progression of arteriosclerotic lesions by TNF-α in VSMCs is well known from several researches. However, the stimulatory effect and related pathways of GA that enhance the expression of adhesion molecules in arteriosclerosis are unknown. These data demonstrated that the expression of molecules through MAPK signaling is further advanced by GA in VSMCs.

### Effects of GA on ROS production and PI3K-AKT activation in TNF-α-stimulated VSMCs

Next, we evaluated the effect of GA on ROS production when an inflammatory reaction occurs in VSMCs. Atherosclerosis is well known as a chronic inflammatory disease and its lesion is exacerbated by ROS production in inflammatory reactions. VSMCs were stimulated with different concentrations of GA (25–50 μM) in the presence of TNF-α (10 ng/mL). GA significantly induced ROS production in TNF-α-induced VSMCs in a concentration-dependent manner (Fig 5A). We also confirmed the effect of GA on ROS production using immunofluorescence microscopy (Fig 5B). At the highest concentration of GA treatment (50 μM), ROS level was increased by approximately three times. On the other hand, it was confirmed that ROS production was suppressed when GA-treated cells were incubated with AG (1 mM). In addition, GA-induced ROS production was closely related to the activation of NF-κB

(A)

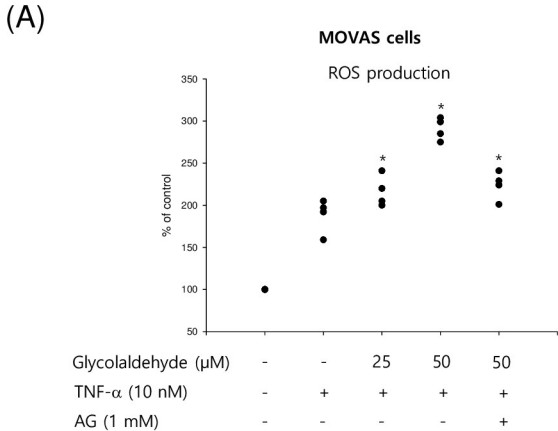

(B)

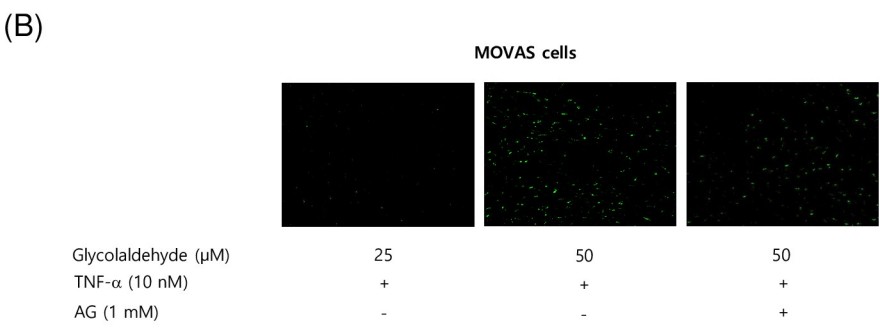

(C) (D)

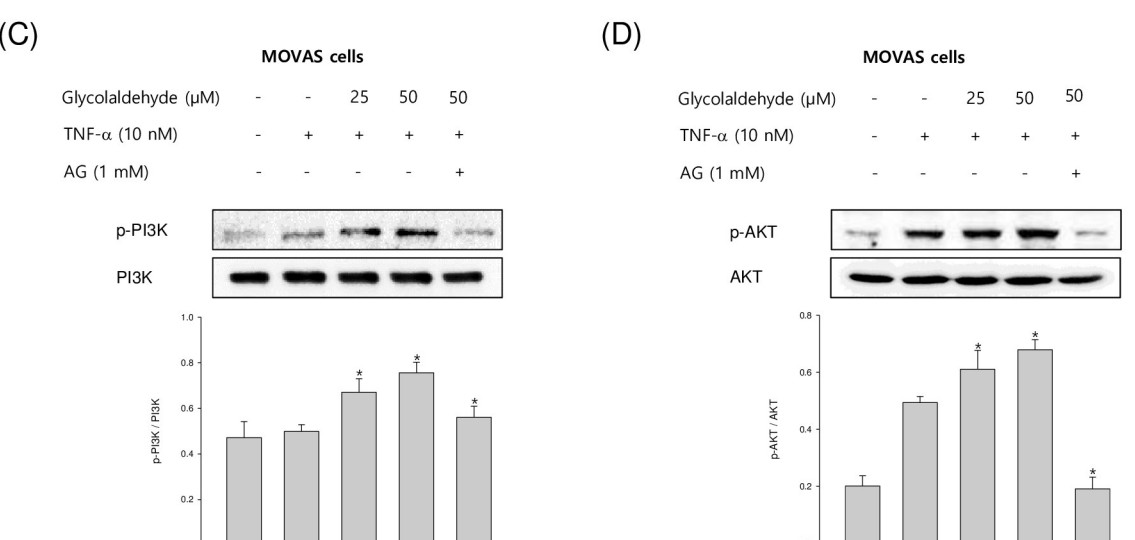

**Fig 5. Effects of GA on the production of ROS and PI3K-AKT activation in TNF-α-stimulated VSMCs.** (A) MOVAS-1 cells were incubated with TNF-α (10 ng/mL) in the presence or absence of GA (25 and 50 μM) for 2 h. The level of ROS was determined as described in materials and methods. (B) The production of ROS in the VSMCs was evaluated using the immunofluorescence microscopy. (C and D) MOVAS-1 cells were activated with TNF-α (10 ng/mL) in the presence or absence of GA (25 and 50 μM) for 4 h. Protein level of PI3K and AKT was determined by Western blot assay. The level of β-actin was measured as an internal control. Results are shown as means ± SEM from a representative experiment (n = 5). $^{*}$p<0.05 significantly different from the group treated with TNF-α.

according to IKB degradation and phosphorylation of MAPKs. Because it was confirmed that ROS generation in vascular cells activates NF-κB and GA treatment further promotes phosphorylation of MAPKs and NF-κB through the production of ROS. To examine whether GA modulated PI3K-AKT signaling pathway in TNF-α-induced MOVAS-1 cells, the cells were treated with or without different concentrations of GA in the presence of TNF-α (10 ng/mL). GA treatment increased the phosphorylation of PI3k and AKT in TNF-α-stimulated VSMCs (Fig 5C and 5D). These finding indicated that GA has a critical role in upregulating PI3K and AKT expression.

### Effect of GA on cytokine production in TNF-α-stimulated VSMCs

We next examined whether GA influenced the release of pro-inflammatory cytokines in TNF-α-activated VSMCs. As shown in Fig 6A and 6B, TNF-α markedly upregulated inflammatory expression. GA further increased the production of TNF-α and IL-6 in a concentration-dependent manner. Moreover, we investigated whether GA could affect pro-inflammatory cytokine expression at the transcriptional level using qRT-PCR analysis. The mRNA expression levels of TNF-α and IL-6 were higher in GA-treated cells compared to control cells (Fig 6 C and 6D). These results correlated with the upregulation of protein expressions, suggesting that GA modulates production of pro-inflammatory cytokines at both protein and mRNA levels in stimulated VSMCs. Collectively, these findings suggested that GA modulates inflammatory responses upregulating the production of pro-inflammatory cytokines.

## Discussion

In this study, we demonstrated that treatment of GA, an AGE precursor, elevated the expression of adhesion molecules VCAM-1 and ICAM-1 in stimulated VSMCs. In addition, GA treatment upregulated the expression of CAM proteins through MAPK/NF-κB and PI3K/AKT signaling pathways. Regulation of CAM proteins VCAM-1 and ICAM-1 will require a critical strategy to prevent and regulate chronic inflammatory disorders including atherosclerosis. To the best of our knowledge, this study is the first to report that GA, a precursor of AGE, affects atherosclerosis by regulating adhesion molecule expression in VSMCs [25, 26]. Concentrations of GA in normal or diseased organisms have not been quantified till date; however most researches have estimated the physiological concentrations which range from 0.1 to 1 mM [27]. Additionally, other AGE precursors similar to GA, such as glyceraldehyde and glyoxal, are also being used at similar concentrations [28, 29]. Therefore, we examined the effect of GA at those concentrations. Quantification of plasma GA level is a crucial important factor in organisms. However, it has not been studied in previous studies. Accordingly, we are planning to proceed with animal experiments for novel finding.

The expression of pro-inflammatory cytokines, such as TNF-α and IL-6, might contribute to adhesion molecule expression in atherosclerotic lesions, which have been found to induce the expression of CAM proteins VCAM-1 and ICAM-1 in VSMCs [2, 9, 30]. Moreover, TNF-α stimulates RAGE expression [20, 31]. Based on these findings, we hypothesized that the AGE-RAGE axis may be a critical intermediate signaling factor to induce inflammatory responses through TNF-α in VSMCs.

Progression of the AGE-RAGE axis causes inflammation and exacerbates the lesion in atherosclerosis. In addition, atherosclerosis is advanced through various signaling systems stimulated by chronic inflammation [10, 32]. It is an important biochemical abnormality accompanying inflammation in the development of atherosclerosis, which plays a critical role. AGE production is a deleterious factor as it not only induces atherosclerotic disease by binding with and activating RAGE in vascular cells but also modifies proteins such as extracellular

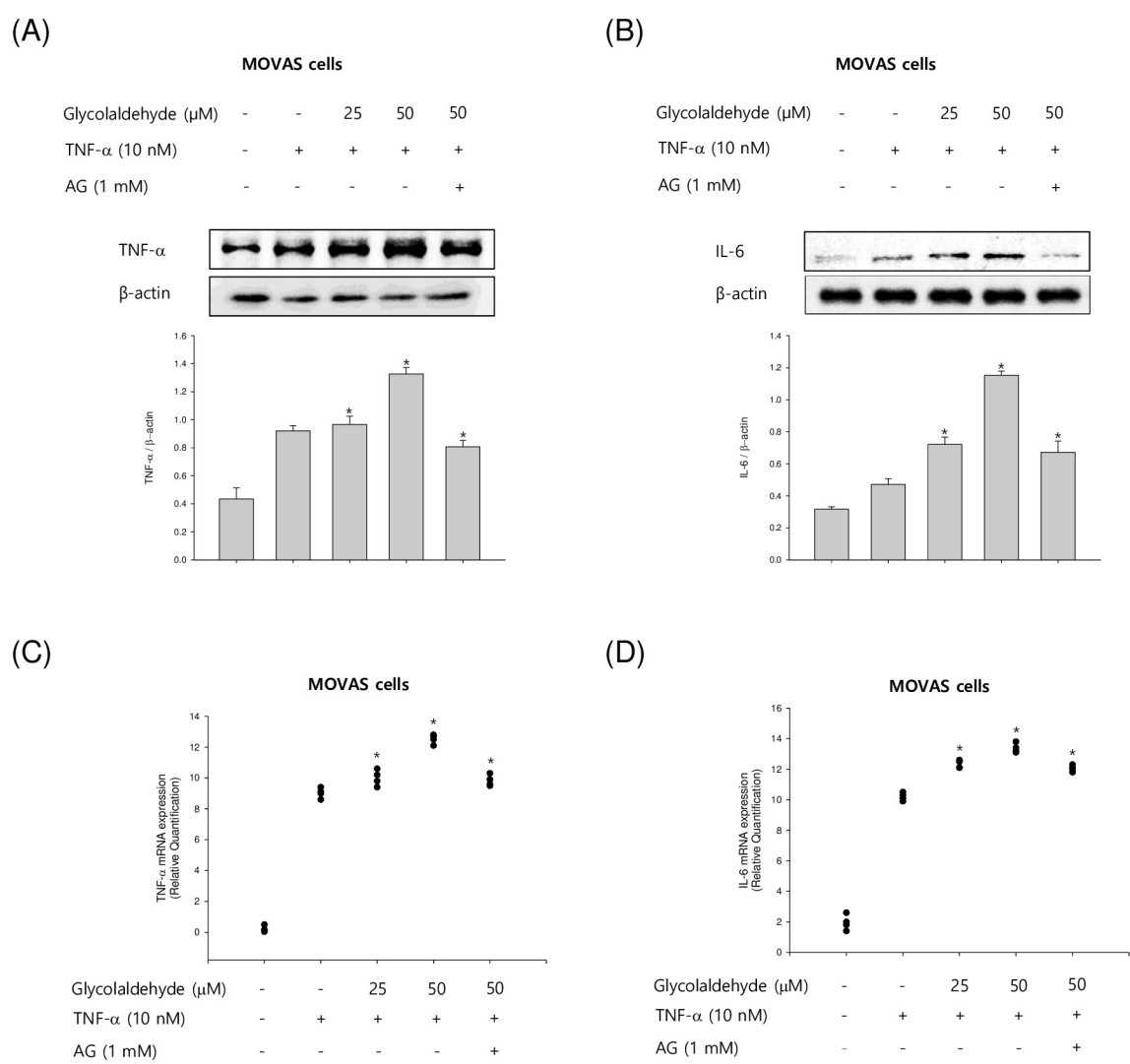

**Fig 6. Effects of GA on the expression of pro-inflammatory cytokines in TNF-α-stimulated VSMCs.** (A and B) The mouse VSMCs, MOVAS-1 cells, were stimulated with TNF-α (10 ng/mL) in the presence or absence of GA (25 and 50 μM) for 4 h. The protein levels of TNF-α and IL-6 were measured by western blot assay. (C and D) The mRNA expression of TNF-α and IL-6 was measured by qRT-PCR. Results are shown as means ± SEM from a representative experiment (n = 5). *p<0.05 significantly different from the group treated with TNF-α.

matrix and circulating lipoproteins [6, 25, 33]. The AGE–RAGE interaction affects cellular signaling, promotes inflammatory mediator expression, and enhances pro-inflammatory cytokines secretion. Genetic manipulation and pharmacological inhibition of the AGE-RAGE pathway showed that the AGE-RAGE pathway is essential in inflammatory responses, specifically in vascular complications [12, 26, 31, 34]. Recent researches have revealed the latent roles of RAGE in the pathogenesis of atherosclerosis. VSMCs exhibit elevated expression of RAGE, which upon interaction with its ligands, increase the production of pro-inflammatory cytokines and CAM proteins [35–37]. Taken together, RAGE may act a pivotal role in vascular diseases by activating inflammation.

Stimulation of RAGE is also known to be related to ROS production, NF-κB activation, as well as recruitment of pro-inflammatory cells. Moreover, RAGE activation is involved in activation of myriads of diverse signaling pathways such as the MAPK, PI3K-AKT, and JAK/

STAT pathways [20, 38, 39]. Additionally, it has been known that cells may regulate the expression of adhesion molecules in VSMCs via the MAPK and NF-κB signaling pathways. Therefore, the phosphorylation of MAPK and NF-κB has an essential role in the regulation of the inflammatory response in vascular disease. Additionally, it is important to phosphorylate and degrade IκB for activation of NF-κB [40–42]. This study demonstrated that GA to regulates cell adhesion molecules expressed through the AGE-RAGE axis via the MAPK/NF-κB signaling pathway in VSMCs [7]. We determined that GA remarkably induced the phosphorylation of MAPKs, and the activation of NF-κB in activated VSMCs, suggesting that GA treatment increased TNF-α-induced VCAM-1 and ICAM-1 expression through the MAPK/NF-κB signaling pathways.

A number of cytokine cause an increase in ROS levels in VSMCs. Pro-inflammatory cytokines, such as TNF-α and IL-6, increase production inflammatory responses. Moreover, increasing evidence indicates that ROS is related to the mechanism of atherosclerosis progression [9, 26, 43]. In this study, treatment of GA remarkably increased ROS production in TNF-α-stimulated VSMCs. It has been indicated that ROS activate several transcriptional factors in VSMCs and may function as a pivotal factor in inflammatory signals that trigger MAPK/NF-κB and PI3K/AKT pathway activation [10]. Furthermore, ROS may to activate the expression of VCAM-1 and ICAM-1 through activation of the NF-κB activation, strongly indicating a possible connection between ROS production and NF-κB signaling. Certainly, these data have indicated that GA-induced NF-κB activity stimulated the TNF-α-stimulated CAM protein expression. Therefore, the effect of GA is due to the adhesion molecules produced by the activation of the MAPK/NF-kB and PI3K/AKT signaling pathways in vascular cells.

In summary, GA stimulated adhesion molecules expression in TNF-α-activated VSMCs. These data indicated a detrimental effect of AGEs in VSMCs. In addition, the effect of GA was mediated by ROS production, phosphorylation of MAPK/NF-κB, and activation of PI3K-AKT. Therefore, our findings have identified GA as a potential detrimental factor in progression of atherosclerosis.

## Supporting information

**S1 Graphical abstract.**
(PDF)

**S1 Raw images.**
(PDF)

## Author Contributions

**Conceptualization:** Hee-Weon Lee, Yoonsook Kim, Sang Keun Ha.

**Data curation:** Hee-Weon Lee.

**Formal analysis:** Hee-Weon Lee, Min Ji Gu, In-Wook Choi, Sang Keun Ha.

**Funding acquisition:** Sang Keun Ha.

**Investigation:** Hee-Weon Lee, Min Ji Gu, In-Wook Choi, Yoonsook Kim, Sang Keun Ha.

**Methodology:** Hee-Weon Lee, Guijae Yoo, Sang-Hoon Lee.

**Project administration:** Yoonsook Kim, Sang Keun Ha.

**Validation:** Hee-Weon Lee, Min Ji Gu, Guijae Yoo.

**Visualization:** Hee-Weon Lee, Min Ji Gu.

Writing – original draft: Hee-Weon Lee, Sang Keun Ha.

Writing – review & editing: Hee-Weon Lee, Sang Keun Ha.

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
