## [Decision Letter · Decision Letter 0]

26 Jul 2021

PONE-D-21-14321

Glycolaldehyde induces vascular inflammation in TNF-α-stimulated vascular smooth muscle cells

PLOS ONE

Dear Dr. Sang Keun Ha

Thank you for submitting your manuscript to PLOS ONE. After careful consideration, we feel that it has merit but does not fully meet PLOS ONE’s publication criteria as it currently stands. Therefore, we invite you to submit a revised version of the manuscript that addresses the all points raised by the reviewers during the review process.

Especially, purity of nuclear and cytosol fractionation should be shown by including the specific nuclear and cytosolic markers in both fractions. Efffects of GA only should be included. Manuscripts have several mistakes and erros, which should be corrected.

From Editor: This study only uses mouse VSMC line, MOVAS-1, which may not reflect intact phenotype.Some key data should be repeated using primary VSMC. In additon, all the averaged graphs should be expressed by dot plots reflecting the individual raw value.

We look forward to receiving your revised manuscript.

Kind regards,

Masuko Ushio-Fukai, PhD

Academic Editor

PLOS ONE

Journal Requirements: 

"This research was supported by the Main Research Program (E 0210200) of the Korea Food Research Institute (KFRI) funded by the Ministry of Science and National Research Foundation of Korea (NRF) grant funded by the Korea government(MSIT) (NRF-2020R1A2C2012608)"

"This research was supported by the Main Research Program (E 0210200) of the Korea Food Research Institute (KFRI) funded by the Ministry of Science and National Research Foundation of Korea (NRF) grant funded by the Korea government(MSIT) (NRF-2020R1A2C2012608).

The fundes have role in conceptualization, formal analysis, investigation, project administration, and writing of the manuscript."

In your cover letter, please note whether your blot/gel image data are in Supporting Information or posted at a public data repository, provide the repository URL if relevant, and provide specific details as to which raw blot/gel images, if any, are not available. Email us at plosone@plos.org if you have any questions

Reviewers' comments:

Reviewer's Responses to Questions

**Comments to the Author**

1. Is the manuscript technically sound, and do the data support the conclusions?

Reviewer #1: Partly

Reviewer #2: No

2. Has the statistical analysis been performed appropriately and rigorously? 

Reviewer #1: No

Reviewer #2: No

3. Have the authors made all data underlying the findings in their manuscript fully available?

Reviewer #1: Yes

Reviewer #2: Yes

4. Is the manuscript presented in an intelligible fashion and written in standard English?

Reviewer #1: No

Reviewer #2: Yes

5. Review Comments to the Author

Reviewer #1: Lee et al found that GA further increased TNFα-induced inflammatory gene expression in mouse VSMC and concluded that GA may contribute atherosclerosis development. This study is not well presented and overstated statements with limited data.

There are several intermediates for AGE formation such as dicarbonyl compounds, such as methylglyoxal, glyoxal, 3-deoxyglucosone as well as GA. What is the rationale to use GA, not other intermediate, in this study?. How GA links with atherosclerosis development?

In introduction, authors described that previous studies reported that deposition of AGE is involved in atherosclerosis via increasing inflammation (Ref 15). Then it is highly expected that AGE precursor, GA involves in inflammation during atherosclerosis. Authors should explain what is new in this study in introduction section.

Reference is missing in last paragraph of introduction section (Line 77-86).

Give detail about source of GA and aminoguanidine, antibodies (especially AGE, RAGE etc), homogenization buffer etc in method section.

As TNF dose mentioned in figures, how authors made 10 nM concentration of TNF.

In Fig 3A, GA increased p-P65 translocation into nucleus but p-P65 in cytosolic fraction also increased. How authors interpret this data?

What is the procedure for nuclear and cytosolic fractionation?. Authors need to show successful fractionation using markers.

In Fig 4B, recombinant TNF treatment decreased protein expression of TNF by SP600125. What is the mechanism?

Provide replication of each experiment in figure legend.

There are several mistakes and missing reference throughout the manuscript.

Reviewer #2: According to previous literature the role of Advanced glycation end products (AGEs) which binds to the Receptor for advanced glycation end products (RAGE) to stimulate the signaling pathways involving in various cellular responses has been established. Indeed in investigating the mechanisms and effects of action of the AGE precursor glycolaldehyde (GA) in inflammation is interesting.

In abstract author claims that “This study investigated the effect of GA on the expression of adhesion molecules in the mouse VSMC line, MOVAS-1”. But throughout the figures author did not show any data related to the treatment of GA only. Instead, all data are reflected with the presence of TNF alpha stimulation. TNF alpha is a well known pro-inflammatory reagent to induce CAM proteins and MAPK signaling. Therefore, the effect of GA on the inflammation is still in question which is the aim of this study.

In addition, the effect of AG inhibitor is not convincing, mainly the Fig 3C, P65 nuclear translocations. Author should provide the good IF images with high resolution to confirm the role of GA on inflammation.

In Fig 1A, TNF-α treatment have no effect on VCAM1 expression which is not acceptable.

In Fig 3A author should confirm the purity of nuclear and cytosolic fractionation by their respective marker proteins. Actin is not accepted as the loading control for this blot.

In Fig 3B, why p-IkBα protein expression is high in basal level while total IkBα protein have no expression in basal.

In Fig 4B, there is a discrepancy in labelling, western blot labelled as TNFα while the respective quantification in bar graph represents IL-1b.

Overall, this manuscript does not fit for publication in PLoS One as it currently stands.

6. PLOS authors have the option to publish the peer review history of their article (what does this mean?). If published, this will include your full peer review and any attached files.

Reviewer #1: No

Reviewer #2: No

---

## [Author Response · Author response to Decision Letter 0]

16 Sep 2021

[September 10, 2021]

Masuko Ushio-Fukai

Editor-in-Chief

PLOS ONE

Dear Editor:

I wish to submit our revised manuscript (PONE-D-21-14321) entitled “Glycolaldehyde induces synergistic effects on vascular inflammation in TNF-α-stimulated vascular smooth muscle cells” for publication in PLOS ONE.

We are grateful to the reviewers for their comments and suggestions, which have helped us to improve our manuscript. We have revised the manuscript based on their comments and have provided our point-by-point responses to each of their comments below.

Editor’s comment

1. Especially, purity of nuclear and cytosol fractionation should be shown by including the specific nuclear and cytosolic markers in both fractions. Efffects of GA only should be included. Manuscripts have several mistakes and errors, which should be corrected.

Response: We agree with the editor's opinion. Therefore, we separated the nucleus from the cytoplasm and proceeded with the experiment again and used appropriate markers. In addition, we screened the effect of GA alone before the experiment to confirm the effect of GA on atherosclerosis. We confirmed that treatment of GA alone had a significant effect. Therefore, we confirmed the synergistic effect of TNF-α and GA.

2. This study only uses mouse VSMC line, MOVAS-1, which may not reflect intact phenotype. Some key data should be repeated using primary VSMC. In additon, all the averaged graphs should be expressed by dot plots reflecting the individual raw value. 

Response: Based on the editor's opinion, we experimented with the primary cell HS-VSMC for key results. Also, we replaced all average graphs by expressing them as point plots. (Fig. 1E and F)

Reviewer 1’s comment

1. Lee et al found that GA further increased TNFα-induced inflammatory gene expression in mouse VSMC and concluded that GA may contribute atherosclerosis development. This study is not well presented and overstated statements with limited data.

Response: We agree with reviewer’s comments. Therefore, we have explained and modified our results within the scope of our study without exaggeration more explicit. (Page 3, Lines 64-65 / Page 10, Lines 232 / Page 11, Lines 246 / Page 12, Lines 280 / Page 13, Lines 285 / Page 15, Lines 352)

2. There are several intermediates for AGE formation such as dicarbonyl compounds, such as methylglyoxal, glyoxal, 3-deoxyglucosone as well as GA. What is the rationale to use GA, not other intermediate, in this study? How GA links with atherosclerosis development?

Response: It is really an important factor. In many studies, many studies have been conducted on the occurrence and development of diseases caused by AGEs. According to several studies, various precursors are known to affect blood vessels by producing AGEs. In addition, AGEs play a role in exacerbating diseases due to excessive oxidative stress and inflammatory responses. It was confirmed that GA also induces this response in cells. We determined that various precursors may play a role based on the study of the effects of AGEs. Therefore, various types of precursors were used to evaluate their effects on VSMCs. As a result, it was confirmed that GA had the greatest effect in VSMCs.

Therefore, we conducted this study using GA on atherosclerosis. This study shows for the first time the acute effects of precursors due to vascular oxidative stress and inflammatory responses, which may provide a better understanding of the pathogenesis of atherosclerosis.

3. In introduction, authors described that previous studies reported that deposition of AGE is involved in atherosclerosis via increasing inflammation (Ref 15). Then it is highly expected that AGE precursor, GA involves in inflammation during atherosclerosis. Authors should explain what is new in this study in introduction section.

Response: In response to comments from the reviewers, we added a note on the relationship between GA and inflammation in the Introduction section. (Page 4, Lines 76-85 / Page 4, Lines 96 - Page 5, Lines 100)

4. Reference is missing in last paragraph of introduction section (Line 77-86).

Response: We have attached references to Lines 77-86 in the Introduction section. (Page 4, Lines 86-93)

5. Give detail about source of GA and aminoguanidine, antibodies (especially AGE, RAGE etc), homogenization buffer etc in method section.

Response: We have added sources and information such as chemicals, antibodies and buffers in materials and methods section. (Page 6, Lines 123-124 / Page 6, Lines 130-136 / Page 7, Lines 165- Page 8, Lines 168)

6. As TNF dose mentioned in figures, how authors made 10 nM concentration of TNF-α.

Response: We selected a concentration of 10 nM to induce inflammation for the atherosclerotic environment in VSMC, known through many previous studies. (Page 7, Lines 145-146)

7. In Fig 3A, GA increased p-P65 translocation into nucleus but p-P65 in cytosolic fraction also increased. How authors interpret this data?

Response: We used beta-actin as a loading control in our existing data. Therefore, it was not possible to know exactly whether the nucleus and cytoplasm were properly separated. We separated the nucleus and cytoplasm again and conducted the experiment again using precise control of the nucleus and cytoplasm, and added accurate results. In the previous study, the phosphorylation form of p65 was increased in the nucleus and cytosol in the TNF-a-induced atherosclerosis model. Our results showed that p65 phosphorylation was increased by GA in the nucleus and cytoplasm. These results were similar to those reported in several previous studies. (Fig. 3A and B)

References

- Kuzhuvelil B. Harikumar, Bokyung Sung, Manoj K. Pandey, Sushovan Guha, Sunil Krishnan and Bharat B. Aggarwal. 2010, 77 (5) 818-827

- Simon Gerhardt, Veronika König, Monika Doll, Tsige Hailemariam-Jahn, Igor Hrgovic, Nadja Zöller, Roland Kaufmann, Stefan Kippenberger and Markus Meissner. 2015, 12-49

8. What is the procedure for nuclear and cytosolic fractionation? Authors need to show successful fractionation using markers.

Response: We have added a method for isolating the cytosol and nucleus of cells to the materials and methods section. In addition, cytosol and nuclear markers were identified and used through re-experiment. (Page 8, Lines 176-183 / Fig. 3A)

9. In Fig 4B, recombinant TNF treatment decreased protein expression of TNF by SP600125. What is the mechanism?

Provide replication of each experiment in figure legend.

Response: In many previous studies, there have been studies that MAPK is involved in TNF-α-induced VSMCs inflammation and the mechanism of adhesion protein development. Among them, JNK plays a key role in this mechanism. Therefore, we confirmed the effect of MAPK, which plays a key role in the inflammatory response in various immune cells as well as in VSMCs, and determined the effect of each MAPK. In addition, we have provided the replication of each experiment in the figure legend following comments from the reviewers. (Page 25, Lines 578-579 / Page 25, Lines 586-587 / Page 26, Lines 596-598 / Page 26, Lines 606-607 / Page 26, Lines 616-617 / Page 27, Lines 624-625)

10. There are several mistakes and missing reference throughout the manuscript.

Response: We reviewed and corrected the mistakes of the manuscript as a whole according to the opinions of the reviewers. We also added missing references. (Page 4, Lines 76-93 / Manuscript)

Reviewer 2’s comment

1. In abstract author claims that “This study investigated the effect of GA on the expression of adhesion molecules in the mouse VSMC line, MOVAS-1”. But throughout the figures author did not show any data related to the treatment of GA only. Instead, all data are reflected with the presence of TNF alpha stimulation. TNF alpha is well known pro-inflammatory reagent to induce CAM proteins and MAPK signaling. Therefore, the effect of GA on the inflammation is still in question which is the aim of this study.

Response: We understand the comments of the reviewers well. We also initially confirmed the effect of GA alone treatment. However, we confirmed that GA alone had a significant effect. Therefore, we confirmed the synergistic effect of GA by creating an atherosclerotic environment in cells through TNF-α. In addition, we use these data to demonstrate that GA has a synergistic effect on TNF-α-induced atherosclerosis.

2. In addition, the effect of AG inhibitor is not convincing, mainly the Fig 3C, P65 nuclear translocations. Author should provide the good IF images with high resolution to confirm the role of GA on inflammation.

Response: We provide high-resolution IF images based on comments from reviewer. Please let us know if you have additional requests for correction of IF images. We will make further corrections. (Fig. 3C)

3. In Fig 1A, TNF-α treatment have no effect on VCAM1 expression which is not acceptable.

Response: We confirmed the effect of TNF-α through several screening procedures. Therefore, a result with a clear effect by GA was selected. However, we agree with the reviewer. Thus, we corrected the data for results in which the effect of TNF-α treatment was clearly visible. (Fig 1A)

4. In Fig 3A author should confirm the purity of nuclear and cytosolic fractionation by their respective marker proteins. Actin is not accepted as the loading control for this blot.

Response: We confirmed and replaced the cytoplasmic and nuclear markers through re-experiment according to the opinions of the reviewers. (Fig 3A)

5. In Fig 3B, why p-IkBα protein expression is high in basal level while total IkBα protein have no expression in basal.

Response: We understand the opinions of our reviewers. We used this band to show the trend of the WB results. We used beta-actin as a loading control in our existing data. Therefore, it was not possible to know exactly whether the nucleus and cytoplasm were properly separated. We separated the nucleus and cytoplasm again and conducted the experiment again using precise control of the nucleus and cytoplasm, and added accurate results. As the reviewer said, we conducted a re-experiment to confirm the exact expression of IkBa and modified it to a more accurate result. (Fig 3B)

6. In Fig 4B, there is a discrepancy in labelling, western blot labelled as TNF-α while the respective quantification in bar graph represents IL-1β.

Response: We have corrected the discrepancy between the picture and the cover. (Fig 4B)

We hope that the questions raised by the editors have been adequately addressed and appreciate your prompt attention to this manuscript. We are looking forward to having this paper published in PLOS ONE.

Sincerely,

Sang Keun Ha

Korea Food Research Institute

245, Nongsaengmyeong-ro, Iseo-myeon

Wanju-gun, Jeollabuk-do 55365

Republic of Korea

Phone: +82-63-219-9358

E mail: skha@kfri.re.kr

---

## [Decision Letter · Decision Letter 1]

4 Jan 2022

PONE-D-21-14321R1Glycolaldehyde induces synergistic effects on vascular inflammation in TNF-α-stimulated vascular smooth muscle cellsPLOS ONE

Dear Dr. Sang Keun Ha

Thank you for submitting your revised manuscript to PLOS ONE. After careful consideration, we feel that it has merit but does not fully meet PLOS ONE’s publication criteria as it currently stands. Therefore, we invite you to submit a revised version of the manuscript that addresses the points raised by 2 reviewers during the review process. Especially, representative blots and cell fractionation assays should be revised.

We look forward to receiving your revised manuscript.

Kind regards,

Masuko Ushio-Fukai, PhD

Academic Editor

PLOS ONE

Journal Requirements:

Reviewers' comments:

Reviewer's Responses to Questions

**Comments to the Author**

1. If the authors have adequately addressed your comments raised in a previous round of review and you feel that this manuscript is now acceptable for publication, you may indicate that here to bypass the “Comments to the Author” section, enter your conflict of interest statement in the “Confidential to Editor” section, and submit your "Accept" recommendation.

Reviewer #1: (No Response)

Reviewer #2: (No Response)

2. Is the manuscript technically sound, and do the data support the conclusions?

Reviewer #1: Partly

Reviewer #2: (No Response)

3. Has the statistical analysis been performed appropriately and rigorously? 

Reviewer #1: No

Reviewer #2: (No Response)

4. Have the authors made all data underlying the findings in their manuscript fully available?

Reviewer #1: (No Response)

Reviewer #2: (No Response)

5. Is the manuscript presented in an intelligible fashion and written in standard English?

Reviewer #1: Yes

Reviewer #2: (No Response)

6. Review Comments to the Author

Reviewer #1: Authors revised manuscript based on reviewer’s comment, but several responses are not satisfactory.

For subcellular fraction (cytosolic and nuclear) assay, the procedure is not well described. What is meaning of buffer A, buffer B. Which kit used for this experiment. In figure 3A, what is the expression level of tubulin in NE and what is the level of Lamin A in CE. This will give the purity of fractions.

What software used for statistical analysis. Which test used for ANOVA.

In response to replication of experiments, Authors mentioned “quintuplicates” in figure legend. This will confuse the readers. Authors should follow the journal format (such as N=5).

Several representative blots are not good (Fig 2B; Fig 4A; Fig 6B), Replace with representative blots.

Reviewer #2: Thank you for your revised manuscript. The current manuscript has improved a lot. Still, I have some minor concerns on the following issues.

In the section of method and materials “Cytosol and nuclear extract preparation” protocol needs detailed information about the kit (like Company name and catalog number) or what is the composition of “buffer A” and “buffer B”.

Fig 2B immunoblot images are not acceptable for publication. Image’s qualities need to be improved.

In Fig 4B immunoblot, 3rd lane and 5th lane labelling are same, but the TNFα and VCAM1 expression in 3rd and 5th lane are completely different. Author should re-check the labelling and the quantification graph as well.

Overall, this manuscript does not fit for publication in PLoS One as it currently stands.

7. PLOS authors have the option to publish the peer review history of their article (what does this mean?). If published, this will include your full peer review and any attached files.

Reviewer #1: No

Reviewer #2: No

---

## [Author Response · Author response to Decision Letter 1]

16 Mar 2022

[March 4, 2022]

Jouranl office

PLOS ONE

Dear Editor:

I wish to submit our revised manuscript (PONE-D-21-14321) entitled “Glycolaldehyde induces synergistic effects on vascular inflammation in TNF-α-stimulated vascular smooth muscle cells” for publication in PLOS ONE.

We are grateful to the reviewers for their comments and suggestions, which have helped us to improve our manuscript. We have revised the manuscript based on their comments and have provided our point-by-point responses to each of their comments below. 

Reviewer 1’s comment

1. For subcellular fraction (cytosolic and nuclear) assay, the procedure is not well described. What is meaning of buffer A, buffer B. Which kit used for this experiment. In figure 3A, what is the expression level of tubulin in NE and what is the level of Lamin A in CE. This will give the purity of fractions.

Response: We added a description of the separation of the nuclear and cytosol. (Page 8, Lines 176-186) In addition, experiments were conducted to confirm the separation of the nucleus and cytoplasm.

2. What software used for statistical analysis. Which test used for ANOVA.

Response: We added the method used for statistical analysis. (Page 9, Lines 206-210)

3. In response to replication of experiments, Authors mentioned “quintuplicates” in figure legend. This will confuse the readers, Authors should follow the journal format (such as N=5).

Response: We modified it to fit the journal format, as mentioned by reviewers. (Page 25, Lines 594; Page 26, Lines 602, 613; Page 27, Lines 622, 632, 640)

4. Several representative blots are not good (Fig 2B; Fig 4A; Fig 6B), Replace with representative blots.

Response: We modified the quality of the bands mentioned. (Fig 2B, Fig 4A, Fig 4B, Fig 5C, Fig 6B)

Reviewer 2’s comment

1. In the section of method and materials “cytosol and nuclear extract preparation” protocol needs detailed information about the kit (like company name and catalog number) or what is the composition of “buffer A” and “buffer B”.

Response: We added information about the kit and composition to the buffer. (Page 8, Lines 176-186)

2. Fig 2B immunoblot images are not acceptable for publication. Image’s qualities need to be improved.

Response: We modified the quality of the bands mentioned. (Fig 2B)

3. In Fig 4B immunoblot, 3rd lane and 5th lane labelling are same, but the TNFa and VCAM1 expression in 3rd and 5th lane are completely different. Author should re-check the labelling and the quantification graph as well.

Response: We checked the bands again and corrected the quantitative graph. (Fig 4B)

We hope that the questions raised by the editors have been adequately addressed and appreciate your prompt attention to this manuscript. We are looking forward to having this paper published in PLOS ONE.

Sincerely,

Sang Keun Ha

Korea Food Research Institute

245, Nongsaengmyeong-ro, Iseo-myeon

Wanju-gun, Jeollabuk-do 55365

Republic of Korea

Phone: +82-63-219-9358

E mail: skha@kfri.re.kr

---

## [Decision Letter · Decision Letter 2]

8 Jun 2022

Glycolaldehyde induces synergistic effects on vascular inflammation in TNF-α-stimulated vascular smooth muscle cells

PONE-D-21-14321R2

Dear Dr. Sang Keun Ha

We’re pleased to inform you that your manuscript has been judged scientifically suitable for publication and will be formally accepted for publication once it meets all outstanding technical requirements.

Kind regards,

Masuko Ushio-Fukai, PhD

Academic Editor

PLOS ONE

Additional Editor Comments (optional):

Reviewers' comments:

Reviewer's Responses to Questions

**Comments to the Author**

1. If the authors have adequately addressed your comments raised in a previous round of review and you feel that this manuscript is now acceptable for publication, you may indicate that here to bypass the “Comments to the Author” section, enter your conflict of interest statement in the “Confidential to Editor” section, and submit your "Accept" recommendation.

Reviewer #1: All comments have been addressed

Reviewer #2: All comments have been addressed

2. Is the manuscript technically sound, and do the data support the conclusions?

Reviewer #1: Yes

Reviewer #2: Yes

3. Has the statistical analysis been performed appropriately and rigorously? 

Reviewer #1: Yes

Reviewer #2: Yes

4. Have the authors made all data underlying the findings in their manuscript fully available?

Reviewer #1: Yes

Reviewer #2: Yes

5. Is the manuscript presented in an intelligible fashion and written in standard English?

Reviewer #1: Yes

Reviewer #2: (No Response)

6. Review Comments to the Author

Reviewer #1: All comments raised by reviewer has been addressed. No other comments

Reviewer #2: (No Response)

7. PLOS authors have the option to publish the peer review history of their article (what does this mean?). If published, this will include your full peer review and any attached files.

Reviewer #1: No

Reviewer #2: No

---

## [Editor Report · Acceptance letter]

24 Jun 2022

PONE-D-21-14321R2 

Glycolaldehyde induces synergistic effects on vascular inflammation
in TNF-α-stimulated vascular smooth muscle cells 

Dear Dr. Ha:

I'm pleased to inform you that your manuscript has been deemed suitable for publication in PLOS ONE. Congratulations! Your manuscript is now with our production department. 

Kind regards, 

on behalf of

Dr. Masuko Ushio-Fukai 

Academic Editor

PLOS ONE